# Looking for Fabry, Finding More: LVH Screening Yields Unexpected Gaucher Diagnosis

**DOI:** 10.3390/medsci13030162

**Published:** 2025-09-01

**Authors:** Sylwia Szczepara, Klaudia Pacia, Katarzyna Trojanowicz, Klaudia Bielecka, Michał Tworek, Zuzanna Sachajko, Katarzyna Holcman, Piotr Podolec, Monika Komar

**Affiliations:** 1Department of Cardiac and Vascular Diseases, Institute of Cardiology, St. John Paul II Hospital, Jagiellonian University Medical College, 31-202 Krakow, Polandk.trojanowicz@szpitaljp2.krakow.pl (K.T.); k.bielecka@szpitaljp2.krakow.pl (K.B.); m.tworek@szpitaljp2.krakow.pl (M.T.); katarzyna.holcman@gmail.com (K.H.); p.podolec@szpitaljp2.krakow.pl (P.P.); 2Department of Cardiac and Vascular Diseases, Institute of Cardiology, St. John Paul II Hospital, 31-202 Krakow, Poland; 3Doctoral School of Medical and Health Sciences, Jagiellonian University Medical College, 31-202 Krakow, Poland; 4Department of Nuclear Medicine, St. John Paul II Hospital, 31-202 Krakow, Poland

**Keywords:** Fabry disease, left ventricular hypertrophy, cardiomyopathy, prevalence study, genetic screening, rare disease

## Abstract

Objective: Fabry disease (FD) is a rare, X-linked lysosomal storage disorder resulting from deficient α-galactosidase A activity, which can manifest as left ventricular hypertrophy (LVH). We aimed to assess the prevalence of FD in an unselected cohort of patients with unexplained LVH. Methods and results: We screened 202 unrelated adults with LVH using enzymatic assays for α-galactosidase A in dried blood spots. Patients with low activity underwent GLA gene sequencing. Echocardiographic parameters were evaluated according to ESC guidelines. FD was diagnosed in 4 women (2%), each carrying distinct pathogenic GLA mutations. All affected individuals showed normal or borderline enzyme activity. Cardiac, renal, or neurological symptoms were observed variably among patients. Echocardiographic findings revealed slightly lower wall thickness and preserved systolic function in FD patients compared to those without FD. Cascade genetic screening identified 16 additional family members with the same mutations. One patient (0.5%) was incidentally diagnosed with Gaucher disease based on syndromic features and enzymatic testing. Conclusions: FD was identified in 2% of patients with unexplained LVH, who were females. Enzyme-based screening followed by targeted genetic testing is a cost-effective strategy for FD detection. Early diagnosis is essential for prompt treatment and family counselling, underscoring the importance of routine FD screening in patients with LVH of unclear aetiology. Our findings support the use of targeted screening for Fabry disease in patients with LVH and systemic features, and highlight the potential to identify other lysosomal disorders in selected cases.

## 1. Introduction

Fabry disease (FD) is a rare X-linked lysosomal storage disorder characterised by a deficiency in α-galactosidase A activity resulting from mutations in the *GLA* gene [1]. The decreased α-galactosidase A activity results in the accumulation of globotriaosylceramide (Gb3), which plays a structural role in cell membranes; however, its aggregation in cells eventually leads to organ failure [2,3]. The analogues and isoforms of Gb3 found in plasma or urine are biomarkers for FD [3]. The disease affects multiple organ systems, including the renal, neurological, ocular, dermatological, and cardiac systems [4]. It manifests in two main forms: the classical form, characterised by early onset, and the atypical form, which typically presents with later onset, often as isolated cardiac or renal complications, particularly in men with residual enzyme activity [5]. It is more difficult to diagnose women as they may have normal enzyme activity levels [6]. Incidence rates vary widely from 1 in 8800 screened newborns to 1 in 170,000 in the overall population [7,8]. As regards the heart, Gb3 is being accumulated in myocardial, valvular, and conduction tissues. This accumulation leads to increased wall thickness, mitral valve thickening, and conduction abnormalities [9]. Damage to myocardial tissue often results in increased left ventricular wall thickness [10]. The prevalence of FD among patients with left ventricular hypertrophy (LVH) is not well understood and remains a topic of debate. Recent studies in various countries suggest that approximately 0.3–1.5% of patients with LVH may have FD, confirmed by genetic testing [5,11,12,13,14]. However, older studies have found prevalence rates as high as 3–6% [15,16].

Patients with isolated cardiac symptoms, such as arrhythmia, chest pain, and diastolic dysfunction, are frequently misdiagnosed with hypertrophic cardiomyopathy (HCM) [17]. Approximately 1000 mutations of the GLA gene have been identified, resulting in multiple forms of FD, from early-onset to asymptomatic types [7]. LVH is typically found in late-onset FD and as such should be an indicator for screening [7,18]. FD is treated with enzyme replacement therapy and novel treatments, such as substrate reduction therapy and mRNA therapy, which are currently being developed [9]. It is imperative that patients with LVH that could be FD carriers receive necessary treatment to prevent the progress of myocardial fibrosis, which ultimately leads to heart failure.

Although Fabry disease is the most well-recognised lysosomal storage disorder associated with LVH, other conditions such as Gaucher disease may present with overlapping clinical features, particularly when accompanied by haematological or visceral abnormalities [19].

The objective of this single-centre study is to evaluate the prevalence of FD in a broad, unselected population of Polish patients with a prior diagnosis of LVH.

## 2. Methods

### 2.1. Study Design

We studied a cohort of consecutive, unrelated patients with left ventricular hypertrophy (LVH) recruited between 2023 and 2024 at the 3rd Referral Hospital in Krakow, Poland. LVH was diagnosed according to the criteria of the World Health Organization (WHO) and the European Society of Cardiology (ESC) Working Group on Myocardial and Pericardial Diseases [20,21]. All patients underwent echocardiography, provided written informed consent for clinical, molecular biology, and genetic studies, and had blood samples collected for further analysis. The study protocol was approved by the institutional ethics committee (approval no. 1072.6120.56.2022, dated 23 March 2022).

### 2.2. Enzymatic and Genetic Screening for Fabry Disease

A simple and sensitive fluorometric method has been developed by an external laboratory for determining α-galactosidase. An assay of dried blood spots on filter paper facilitates the screening of FD in at-risk populations. The assay uses 4-methylumbelliferyl-α-D-galactose as a synthetic substrate for the enzyme. Following incubation, protein precipitation is used to both stop the enzymatic reaction and remove interfering proteins, minimizing the risk of false positives due to fluorescence quenching. Patients with low plasma α-galactosidase A activity (0 to 2.8 μmol/L/h) underwent a genetic study of the *GLA* gene.

The DNA mutations are described according to the *GLA* complementary DNA sequence [22], with the A of the ATG initiation codon being +1. The diagnosis of FD was made in those patients with low enzymatic activity who also potentially had disease-causing mutations in the GLA gene.

In cases where the clinical picture included unexplained anaemia, thrombocytopenia, and/or splenomegaly, additional enzymatic testing for β-glucocerebrosidase activity was performed using the same DBS platform. In patients with low enzyme activity, GBA gene sequencing was conducted to confirm or exclude Gaucher disease.

### 2.3. Echocardiographic Assessment

The examinations were performed using a Philips EPIQ 7C ECHO machine by an experienced, certified echocardiographer. The study adhered to the guidelines of the European Association of Cardiovascular Imaging (EACVI) and the 2023 European Society of Cardiology (ESC) Guidelines for the management of cardiomyopathies [21,23]. Hypertrophic cardiomyopathy (HCM) was defined by an LV wall thickness ≥ 15 mm in any myocardial segment that is not explained solely by loading conditions. Less severe degrees of wall thickening (13–14 mm) required evaluation of other features including family history, genetic findings, and ECG abnormalities [21]. The assessment included both systolic and diastolic function of the left and right ventricles, as well as evaluation of valvular apparatus function. All measurements were conducted using dedicated software provided by the manufacturer.

### 2.4. Magnetic Resonance Imaging

Cardiac magnetic resonance imaging (CMR) in patients with FD was performed according to the 2020 guidelines of the Society for Cardiovascular Magnetic Resonance (SCMR). Cine-SSFP sequences were acquired in two-, three-, and four-chamber views, as well as in the short-axis plane. T1 mapping techniques (MOLLI, ShMOLLI, SASHA) and T2 mapping techniques (GRASE, T2-Prep, SSFP, MESE) were performed in both short- and long-axis planes. The optimal inversion time was determined using the TI scout sequence, and gadolinium contrast administration enabled late gadolinium enhancement (LGE) imaging for the assessment of myocardial fibrosis and extracellular volume. In patients with FD, characteristic findings include papillary muscle hypertrophy, reduced native T1 values, and mid-wall scarring of the inferolateral wall observed on LGE images.

### 2.5. Statistical Analysis

Descriptive statistics were used to summarize the study data. Continuous variables were reported as medians and interquartile ranges (IQRs). All statistical analyses were performed using Microsoft Excel (Microsoft Corporation, Redmond, WA, USA).

## 3. Results

### 3.1. Baseline Characteristics

A total of 202 patients with increased LVH were included in the analysis, among whom 4 were diagnosed with FD (2%). The median age of the overall study population was 47.0 years (IQR: 37.0–58.0), identical to non-FD patients. In contrast, patients with FD were older, with a median age of 55.0 years (IQR: 50.3–56.8). Women comprised 43% of the total cohort and 100% of the FD group (Table 1).

Anthropometric parameters, including height, weight, body surface area (BSA), and body mass index (BMI), were comparable across groups. The median BMI in the total cohort was 27.0 kg/m^2^ (IQR: 23.8–30.3), while FD patients had a median BMI of 27.5 kg/m^2^ (IQR: 25.8–29.0) (Table 1).

The median serum α-galactosidase A activity in the overall cohort was 5.3 μmol/L/h (IQR: 4.4–6.9), slightly higher in the FD subgroup at 6.0 μmol/L/h (IQR: 4.6–6.8). Plasma lyso-Gb3 concentration, measured in a subset of patients (*n* = 129), was higher in FD patients (median: 6.9 nmol/L, IQR: 5.2–8.4) compared to the non-FD group (median: 2.1 nmol/L, IQR: 1.6–2.8) (Table 1).

### 3.2. Characteristics of Patients with Fabry Disease

Among the 4 patients diagnosed with FD, all were women, each carrying a distinct pathogenic GLA gene variant: c.803_806del, c.126G>C (p.Met42Ile), c.1069C>T, and c.484del. Their ages ranged from 36 to 62 years. Enzymatic activity of α-galactosidase A varied between 2.2 and 6.8 μmol/L/h, with the lowest value observed in the youngest patient. Lyso-Gb3 levels ranged from 3.9 to 9.5 nmol/L. Clinical manifestations were heterogeneous, with one patient exhibiting renal manifestations, and one reporting neurological symptoms. All patients developed their first symptoms in childhood, between the ages of 8 and 10. CMR revealed features consistent with the cardiac phenotype of FD in two out of four patients, which include mid-layer posterolateral gadolinium enhancement in one patient and low native T1 in another patient. Further investigations were conducted in the families of the patients diagnosed with FD. Detailed patient characteristics are included in Table 2.

### 3.3. Echocardiographic Characteristics

Left ventricular internal diameters in systole (LVIDs) and diastole (LVIDd) were slightly lower in FD patients compared to the non-FD group [LVIDs: 45.5 mm (36.0–49.0) vs. 47.0 mm (42.0–52.0); LVIDd: 28.0 mm (24.5–31.5) vs. 30.5 mm (25.3–37.0)] (Table 3). Measurements of wall thickness showed similar trends, with FD patients exhibiting slightly lower values in both posterior wall and interventricular septum thickness during systole and diastole. Notably, the median interventricular septum thickness in diastole (IVSd) was 15.5 mm (IQR: 15.0–16.0) in FD patients compared to 18.0 mm (IQR: 16.0–22.5) in the non-FD group (Table 3). Left ventricular ejection fraction (EF) was preserved in both groups, with a higher median value observed in FD patients [65.0% (65.0–65.0) vs. 60.0% (55.0–65.0)]. Right ventricular function as measured by TAPSE was also slightly higher in FD patients [25.0 mm (25.0–30.0)] versus non-FD patients [23.0 mm (20.0–28.0)] (Table 3). There were no marked differences in left atrial and right atrial dimensions or filling pressures between groups. The E/A ratio and E/E′ values in FD patients were within the expected range and comparable to non-FD patients (Table 3). Figure 1 (parasternal long-axis) and Figure 2 (apical four-chamber) present concentric left ventricular hypertrophy consistent with hypertrophic cardiomyopathy in a patient with genetically confirmed Fabry disease (own clinical material).

### 3.4. Genetic Findings and Family Analysis

Figure 3 illustrates the pedigrees of the four families in which index patients (marked in red) were diagnosed with FD. In each family, molecular analysis revealed a distinct variant in the *GLA* gene: a deletion (c.803_806del) in Family 1, a missense mutation (c.126G>C; p.Met42Ile) in Family 2, a nucleotide substitution (c.1069C>T) in Family 3, and another deletion (c.484del) in Family 4.

All identified patients were female and transmitted the pathogenic variant following an X-linked inheritance pattern. Across all families, a total of 16 relatives were found to carry the same pathogenic mutation, including 8 affected males and 8 females. Additionally, individuals marked in grey were considered potential initial carriers within the lineage but had not been genetically confirmed at the time of analysis.

### 3.5. Incidental Finding of Gaucher Disease

In one male patient (0.5%), additional diagnostic workup was initiated due to the coexistence of unexplained concentric LVH with normocytic anaemia (Hb 10.2 g/dL), thrombocytopenia (82 × 10^9^/L), and splenomegaly (18 cm in length on ultrasound). Dried blood spot testing revealed markedly decreased β-glucocerebrosidase activity (<2.3 μmol/L/h), and subsequent GBA gene sequencing confirmed a pathogenic homozygous c.1226A>G (p.Asn409Ser) variant, consistent with type 1 Gaucher disease. Cardiac MRI showed concentric hypertrophy of the left ventricle (IVSd 16 mm, LVEF 61%) without late gadolinium enhancement. Enzyme replacement therapy with imiglucerase was initiated. This incidental diagnosis highlights the broader diagnostic potential of red-flag-driven lysosomal screening in patients with LVH and systemic features.

## 4. Discussion

Our study demonstrates a 2% prevalence of genetically confirmed FD in an unselected cohort of 202 patients with unexplained LVH. All affected individuals in our cohort were women, and all carried distinct pathogenic variants of the *GLA* gene, underscoring both the genetic and phenotypic heterogeneity of the disease. The phenotypic expression of FD in our cohort was heterogeneous, with patients presenting isolated cardiac, renal, or neurological manifestations. Interestingly, echocardiographic measurements revealed relatively modest LV wall thickening in the FD group compared to LVH patients without FD, alongside preserved systolic function. These subtle cardiac features highlight the importance of clinical vigilance and comprehensive evaluation in suspected cases.

In other studies, the reported prevalence of FD among patients with hypertrophic cardiomyopathy is approximately 0.5–1% [22,24]. These findings are consistent across various populations and highlight the need for routine screening in selected cardiac cohorts. In a Spanish cohort of 508 unrelated patients with hypertrophic cardiomyopathy, screening based on plasma α-galactosidase A activity and subsequent GLA gene sequencing identified FD in 5 individuals (1%), including 3 men and 2 women [22]. Familial genetic studies identified 14 additional mutation carriers, highlighting the diagnostic and therapeutic value of targeted screening in this population. In a large, multicentre prospective screening study conducted across 55 centres in China, FD was genetically confirmed in 8 out of 906 patients (0.88%) with unexplained left ventricular hypertrophy (LVH), using dried blood spot testing for α-galactosidase A activity and lyso-Gb3 [25]. Family studies revealed 14 additional mutation carriers, supporting the utility of combining enzymatic and biomarker screening with cascade testing in this population.

Importantly, all four patients with FD exhibited either borderline or within-normal-range α-galactosidase A activity levels, emphasizing a key limitation of enzyme-based screening in female patients. In our cohort, α-galactosidase A activity ranged from 2.2 to 6.8 μmol/L/h. Likewise, lyso-Gb3 concentrations were elevated in all affected individuals, ranging from 3.9 to 9.5 nmol/L, further supporting the diagnosis of FD despite the variable enzymatic activity. In addition to its diagnostic value, lyso-Gb3 may serve as a valuable marker for monitoring disease activity or response to therapy, particularly in patients with atypical or oligosymptomatic presentations. Therefore, while enzymatic assays are suitable for initial male screening, our findings reinforce the necessity of reflex genetic testing in females with borderline enzyme activity and suggestive clinical or family history.

Recent evidence from a systematic review and meta-analysis including 14 studies and over 1700 patients with FD demonstrated that several imaging parameters are significant predictors of adverse clinical outcomes. In particular, late gadolinium enhancement on CMR was most strongly associated with poor prognosis, while other markers such as left atrial volume, LV mass, wall thickness, E/e′ ratio, and global longitudinal strain also correlated with cardiovascular risk. In contrast, LV ejection fraction and T1-mapping showed limited prognostic value. These findings highlight the importance of incorporating advanced imaging markers into risk stratification of FD patients [26].

Recent evidence indicates that enzyme replacement therapy (ERT) in FD is associated with stabilization of key cardiac MRI parameters, including left ventricular mass (LVM), maximum wall thickness (MLVWT), and native T1 values. While small reductions in LVM and MLVWT were observed, late gadolinium enhancement continued to increase, suggesting ongoing myocardial fibrosis despite treatment. A meta-analysis of 11 studies comprising 445 patients therefore supports the view that ERT can slow disease progression but may not completely halt structural remodelling of the myocardium [27].

One of the most significant findings of our study is that the diagnosis of FD in 4 patients has enabled the identification of 16 additional carriers. FD had not been previously diagnosed in any of the families. Although the overall prevalence of FD in this cohort was relatively low, the clinical significance of early diagnosis is considerable. Our findings support the incorporation of FD screening into the routine diagnostic algorithm for patients with unexplained LVH, particularly when no alternative aetiology is identified. Early diagnosis and initiation of disease-specific therapy not only prevent irreversible organ damage but may also improve long-term cardiac outcomes, delay progression to heart failure, and enhance overall survival. These benefits provide a strong rationale for routine screening of LVH patients to identify FD at a stage when treatment is most effective.

Although this study focused on FD, one patient was diagnosed with type 1 Gaucher disease based on systemic red flags. This incidental finding illustrates that red-flag-guided screening may occasionally reveal other lysosomal conditions presenting with LVH and supports a broader diagnostic perspective in selected patients.

This study has several limitations. First, the study was conducted at a single tertiary cardiology centre, which may limit the generalizability of the results to broader populations or primary care settings. Second, the number of patients diagnosed with FD was small (*n* = 4), which precluded subgroup analyses and limited the ability to draw conclusions about genotype–phenotype correlations. Lastly, long-term clinical outcomes, treatment responses, and follow-up data for patients and their relatives identified through cascade screening were not assessed, which limits insight into the impact of early diagnosis and therapeutic intervention.

## 5. Conclusions

In this single-centre study of 202 patients with LVH, we identified FD in 2% of the cohort, all of whom were female. Cascade family screening led to the identification of 16 additional carriers, underlining the value of early diagnosis not only for affected individuals but also for their relatives. Our findings support the inclusion of FD screening in patients with unexplained LVH. A screening strategy based on initial enzymatic testing, followed by *GLA* gene sequencing in selected cases, appears to be cost-effective and clinically meaningful. Early diagnosis of FD enables the timely initiation of disease-specific therapies, such as enzyme replacement or chaperone therapy, which may alter the natural course of the disease and prevent irreversible organ damage. These results further highlight the importance of raising awareness among clinicians about the cardiac manifestations of FD and the potential benefit of systematic genetic screening in appropriate patient populations.

Moreover, these findings also suggest that, in selected cases, such screening may provide diagnostic value beyond FD by uncovering other rare lysosomal conditions with overlapping features.

## Figures and Tables

**Figure 1 medsci-13-00162-f001:**
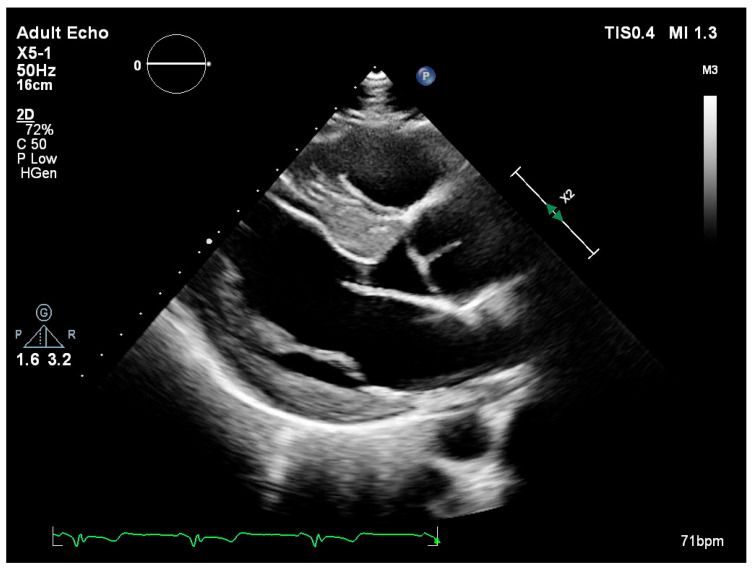
Transthoracic echocardiography—parasternal long-axis view showing severe left ventricular hypertrophy.

**Figure 2 medsci-13-00162-f002:**
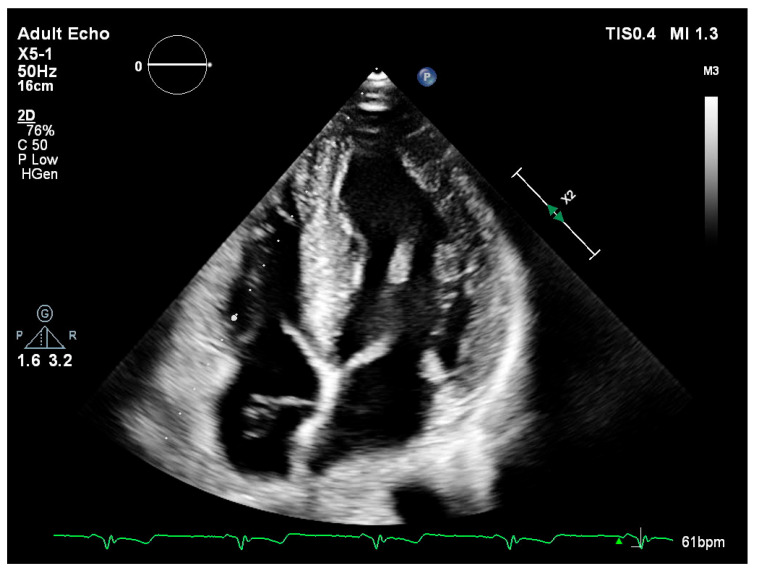
Transthoracic echocardiography—apical four-chamber (A4C) view showing concentric left ventricular hypertrophy.

**Figure 3 medsci-13-00162-f003:**
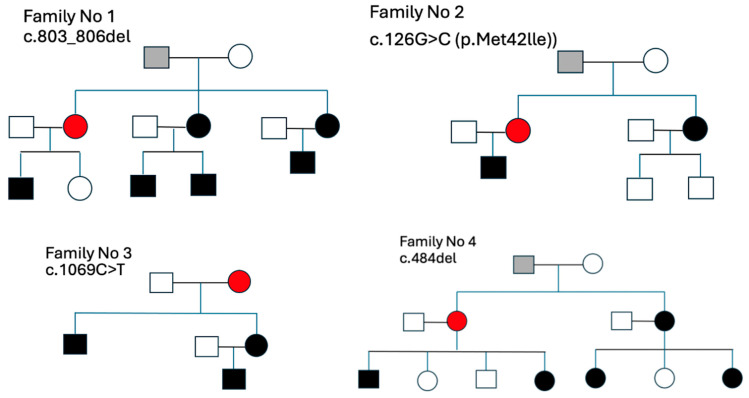
Family pedigrees of patients with Fabry disease. Squares represent males and circles represent females. Black-filled symbols indicate individuals with a confirmed pathogenic mutation, red symbols denote index patients diagnosed with Fabry disease, and grey symbols mark individuals suspected of being the initial carriers within the family lineage.

**Table 1 medsci-13-00162-t001:** Demographic characteristics.

Parameter	All Patients, Median (IQR) (N = 202)	Non-FD Patients, Median (IQR) (N = 198)	FD Patients, Median (IQR) (N = 4)
Age (years)	47.0 (37.0–58.0)	47.0 (37.0–58.0)	55.0 (50.3–56.8)
Women (%)	43%	41%	100%
Height (m)	1.7 (1.7–1.8)	1.7 (1.7–1.8)	1.7 (1.7–1.7)
Weight (kg)	82.0 (70.0–91.0)	82.0 (70.0)	77.5 (69.0–81.0)
BSA (m^2^)	1.9 (1.8–2.1)	1.9 (1.8–2.1)	1.9 (1.8–1.9)
BMI (kg/m^2^)	27.0 (23.8–30.3)	27 (23.8–30.2)	27.5 (25.8–29.0)
Alpha-galactosidase (μmol/L/h)	5.3 (4.4–6.9)	5.3 (4.4–7.0)	6.0 (4.6–6.8)
Lyso-GL3 (nmol/L)	2.1 (1.6–2.8) (*n* = 129)	2.1 (1.6–2.8) (*n* = 125)	6.9 (5.2–8.4)

BMI—body mass index; BSA—body surface area; FD—Fabry disease; IQR—interquartile range; Lyso-GL3—lyso-globotriaosylsphingosine.

**Table 2 medsci-13-00162-t002:** The clinical and genetic characteristics of patients with Fabry disease.

Patient	1	2	3	4
Gender	Female	Female	Female	Female
Age (years)	36	55	62	55
GLA mutation	c.803_806del	c.126G>C (p.Met42Ile)	c.1069C>T	c.484del
Alpha-galactosidase (μmol/L/h in serum)	2.2	6.8	6.8	5.2
Lyso-GL3 (nmol/L)	7.3	9.5	3.9	6.4
First manifestations of the disease	hypohidrosis, acroparesthesias	acroparesthesias, abdominal pain and diarrhea	acroparesthesias, angiokeratoma	angiokeratoma, abdominal pain and constipation, hypohidrosis
Renal manifestation	+	-	-	-
Neurological manifestation	-	-	+	-
Hypertrophy of papillary muscles	-	-	-	-
Mid Layer posterolateral gadolinium enhancement	-	-	-	+
Low native T1	-	+	-	-

GLA—gene encoding α-galactosidase A; Lyso-GL3—lyso-globotriaosylsphingosine; T1—longitudinal relaxation time in cardiac magnetic resonance imaging.

**Table 3 medsci-13-00162-t003:** Echocardiographic characteristics.

Parameter	Non-FD Patients—Median (IQR)	N (Non-FD Patients)	FD Patients—Median (IQR)	N (FD Patients)
LVIDs (mm)	47.0 (42.0–52.0)	143	45.5 (36.0–49.0)	4
LVIDd (mm)	30.5 (25.3–37.0)	138	28.0 (24.5–31.5)	4
PWTs (mm)	14.0 (13.0–15.0)	148	13.0 (13.0–15.0)	4
PWTd (mm)	17.0 (15.0–19.0)	133	16.5 (15.0–23.0)	4
IVSs (mm)	16.0 (13.5–17.5)	148	13.5 (13.0–14.0)	4
IVSd (mm)	18.0 (16.0–22.5)	133	15.5 (15.0–16.0)	4
EF (%)	60.0 (55.0–65.0)	148	65.0 (65.0–65.0)	4
RVOT (mm)	30.0 (27.0–36.0)	140	27.5 (27.0–33.0)	4
TAPSE (mm)	23.0 (20.0–28.0)	130	25.0 (25.0–30.0)	3
RVD1 (mm)	38.0 (33.0–41.0)	98	35.0 (34.0–36.0)	2
LAD (mm)	39.0 (34.0–45.0)	137	37.0 (35.0–52.0)	4
LAA (mm^2^)	23.0 (17.0–27.0)	135	18.5 (16.5–30.0)	4
RAD (mm)	38.0 (33.0–47.5)	19	-	0
RAA (mm^2^)	19.0 (15.0–25.5)	136	18.0 (14.5–26.0)	3
E/A ratio	1.0 (0.8–1.4)	133	1.1 (0.9–1.2)	2
E/E’	9.0 (6.7–12.0)	118	10.1 (6.2–12.8)	3
CVP (mmHg)	3.0 (3.0–8.0)	117	3.0 (3.0–3.0)	3
IVC (mm)	18.0 (15.0–21.0)	120	16.0 (15.0–19.0)	3

LVIDs—left ventricular internal diameter in systole, LVIDd—left ventricular internal diameter in diastole, PWTs—posterior wall thickness in systole, PWTd—posterior wall thickness in diastole, IVSs—interventricular septal thickness in systole, IVSd—interventricular septal thickness in diastole, EF—ejection fraction, RVOT—right ventricular outflow tract, TAPSE—tricuspid annular plane systolic excursion, RVD1—right ventricular diameter at the base, LAD—left atrial diameter, LAA—left atrial area, RAD—right atrial diameter, RAA—right atrial area, E/A ratio—ratio of early to late ventricular filling velocities, E/E′—ratio of early mitral inflow velocity to early diastolic mitral annular velocity, CVP—central venous pressure, IVC—inferior vena cava diameter, and FD—Fabry disease.

## Data Availability

The data that support the findings of this study are available from the corresponding author, upon reasonable request.

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
