# Peer review of "Looking for Fabry, Finding More: LVH Screening Yields Unexpected Gaucher Diagnosis"

_medsci, 2025, doi:10.3390/medsci13030162_

Round 1

Reviewer 1 Report

Comments and Suggestions for Authors

his article reports the findings of a single-center study that screened 202 patients with unexplained left ventricular hypertrophy (LVH) for Fabry disease, a rare X-linked lysosomal storage disorder. The authors used an enzyme-based screening approach followed by genetic testing to identify cases of Fabry disease in this cohort. The study not only provides insights into the prevalence of Fabry disease among LVH patients but also highlights the potential for incidentally detecting other lysosomal disorders through targeted screening.

Comments/Suggestions:

  1. The study design and screening methodology are appropriate for the stated objectives. The use of dried blood spot testing for initial enzyme activity screening, followed by genetic confirmation, is a well-established and cost-effective approach for Fabry disease detection.

  2. The authors should be commended for conducting cascade family screening, which led to the identification of additional Fabry disease carriers. This underscores the importance of early diagnosis for prompt treatment initiation and family counseling.

  3. The incidental finding of Gaucher disease in one patient is an interesting observation and supports the authors' recommendation for a broader diagnostic perspective in selected cases with systemic features suggestive of lysosomal disorders.

  4. The echocardiographic findings in Fabry disease patients, particularly the relatively modest LV wall thickening and preserved systolic function, are noteworthy and highlight the potential for subtle cardiac phenotypes in some cases.

  5. While the overall prevalence of Fabry disease in this cohort (2%) is consistent with previous reports, the small number of affected individuals (n=4) limits the ability to draw robust genotype-phenotype correlations or subgroup analyses.

  6. The authors could consider discussing the potential impact of early diagnosis and treatment on long-term outcomes, as this information could further strengthen the rationale for routine screening in LVH patients.

  7. The study's limitations, including the single-center design and lack of long-term follow-up data, are appropriately acknowledged by the authors.

Author Response

This article reports the findings of a single-center study that screened 202 patients with unexplained left ventricular hypertrophy (LVH) for Fabry disease, a rare X-linked lysosomal storage disorder. The authors used an enzyme-based screening approach followed by genetic testing to identify cases of Fabry disease in this cohort. The study not only provides insights into the prevalence of Fabry disease among LVH patients but also highlights the potential for incidentally detecting other lysosomal disorders through targeted screening.

Authors: Thank you for such a positive review and kind words. We have carefully considered your suggestions and incorporated the proposed changes. Please find our responses below

Comments/Suggestions:

  1. The study design and screening methodology are appropriate for the stated objectives. The use of dried blood spot testing for initial enzyme activity screening, followed by genetic confirmation, is a well-established and cost-effective approach for Fabry disease detection.

Authors: Thank you for this comment.

  1. The authors should be commended for conducting cascade family screening, which led to the identification of additional Fabry disease carriers. This underscores the importance of early diagnosis for prompt treatment initiation and family counseling.

Authors: Thank you for such a nice words.

  1. The incidental finding of Gaucher disease in one patient is an interesting observation and supports the authors' recommendation for a broader diagnostic perspective in selected cases with systemic features suggestive of lysosomal disorders.

Authors: Thank you for this comment.

  1. The echocardiographic findings in Fabry disease patients, particularly the relatively modest LV wall thickening and preserved systolic function, are noteworthy and highlight the potential for subtle cardiac phenotypes in some cases.

Authors: Thank you for this comment.

  1. While the overall prevalence of Fabry disease in this cohort (2%) is consistent with previous reports, the small number of affected individuals (n=4) limits the ability to draw robust genotype-phenotype correlations or subgroup analyses.

Authors: We agree, and stated in the discussion that the number of patients diagnosed with FD was small (n=4), which precluded subgroup analyses and limited the ability to draw conclusions about genotype–phenotype corre-lations.

  1. The authors could consider discussing the potential impact of early diagnosis and treatment on long-term outcomes, as this information could further strengthen the rationale for routine screening in LVH patients.

Authors: Thank you for this valuable comment. We have revised one of the paragraphs in the Discussion section accordingly, and the changes have been highlighted in red.

  1. The study's limitations, including the single-center design and lack of long-term follow-up data, are appropriately acknowledged by the authors.

Authors: Thank you for this comment.

Reviewer 2 Report

Comments and Suggestions for Authors

The manuscript is prepared following the principles of good scientific practice. There are indications of compliance with ethical requirements, and the statistics are accurate. Tables and illustrations are informative, with qualitative explanations. I think conclusions are based on the obtained results. The material has several significant limitations, such as the number of identified patients and the lack of long-term observation data. Still, in the current format, the authors have achieved the goals set in the manuscript. 

Author Response

The manuscript is prepared following the principles of good scientific practice. There are indications of compliance with ethical requirements, and the statistics are accurate. Tables and illustrations are informative, with qualitative explanations. I think conclusions are based on the obtained results. The material has several significant limitations, such as the number of identified patients and the lack of long-term observation data. Still, in the current format, the authors have achieved the goals set in the manuscript.

Authors: Thank you very much for your positive evaluation of our work and for acknowledging the scientific quality of the manuscript. We appreciate your recognition of the study design, statistical accuracy, and clarity of the tables and illustrations. We also agree with the noted limitations and have emphasized them in the Discussion. Your feedback is highly valuable and encouraging for us.

Reviewer 3 Report

Comments and Suggestions for Authors The manuscript addresses the important topic of Fabry disease prevalence among patients with unexplained LVH. The additional value on available literature is genetic screening and the incidental diagnosis of Gaucher disease, with proper cascade familiy studies.    Main issues:  - Very small number of FD cases (n=4), limiting conclusions. - Single-centre design, no longitudinal follow-up. - CMR findings are underreported despite being described in methods. Please clarify rationale for limited use of CMR and expand on imaging findings. - Strengthen discussion with recent systematic reviews on imaging predictors and therapy effects (e.g., Stankowski et al., Eur J Clin Invest 2025; Figliozzi et al., Radiol Cardiothoracic Imaging 2024).

Author Response

The manuscript addresses the important topic of Fabry disease prevalence among patients with unexplained LVH. The additional value on available literature is genetic screening and the incidental diagnosis of Gaucher disease, with proper cascade familiy studies.    Main issues:  - Very small number of FD cases (n=4), limiting conclusions. - Single-centre design, no longitudinal follow-up. - CMR findings are underreported despite being described in methods. Please clarify rationale for limited use of CMR and expand on imaging findings. - Strengthen discussion with recent systematic reviews on imaging predictors and therapy effects (e.g., Stankowski et al., Eur J Clin Invest 2025; Figliozzi et al., Radiol Cardiothoracic Imaging 2024).

Authors: Thank you for your valuable comments. We acknowledge the limitation of a small number of FD cases and the single-centre design, which we have further emphasized in the revised Limitations section. We also recognize the absence of longitudinal follow-up data and note that patients are being monitored for future outcome analyses. Regarding cardiac MRI, we agree that results are limited but we reported all findings in table 2, and in the text: „CMR revealed features consistent with the cardiac phenotype of FD in two out of four patients, which include mid-layer posterolateral gadolinium enhancement in one patient and low native T1 in one patient.” Thank you for the valuable suggestion to include these two important articles. We have now added them to the Discussion section.

Round 2

Reviewer 3 Report

Comments and Suggestions for Authors

The authors did wrong in citing one paper which is off-topic: Figliozzi Set al., Myocardial Fibrosis at Cardiac MRI Helps Predict Adverse Clinical Outcome in Patients with Mitral Valve Prolapse. doi: 10.1148/radiol.220454. Epub 2022 Sep 13. PMID: 36098639.

I suggest the paper is that Figliozzi Set al., Effects of Enzyme Replacement Therapy on Cardiac MRI Findings in Fabry Disease: A Systematic Review and Meta-Analysis. Radiol Cardiothorac Imaging. 2024 Jun;6(3):e230154. doi: 10.1148/ryct.230154. PMID: 38842453; PMCID: PMC11211942. 

Please replace the 27th piece of literature.

On top of that, they improved the paper. 

Author Response

We thank the reviewer for carefully noting the incorrect reference. We agree with the suggestion and have replaced the previously cited article (Figliozzi S et al., Myocardial Fibrosis at Cardiac MRI Helps Predict Adverse Clinical Outcome in Patients with Mitral Valve Prolapse, Radiology, 2022) with the more relevant reference (Figliozzi S et al., Effects of Enzyme Replacement Therapy on Cardiac MRI Findings in Fabry Disease: A Systematic Review and Meta-Analysis, Radiol Cardiothorac Imaging, 2024). This correction has been implemented, and the revised reference now appears as citation number 27 in the manuscript.

We also appreciate the reviewer’s positive feedback on the overall improvement of the paper.